# Quantitative Evaluation of the Reduced Capacity of Skeletal Muscle Hypertrophy after Total Body Irradiation in Relation to Stem/Progenitor Cells

**DOI:** 10.3390/jcm11133735

**Published:** 2022-06-28

**Authors:** Tsuyoshi Fukuzawa, Toshiharu Natsume, Miyu Tamaki, Takeshi Imai, Ippei Yamato, Tetsuro Tamaki

**Affiliations:** 1Department of Radiation Oncology, Tokai University School of Medicine, 143 Shimokasuya, Isehara 259-1193, Kanagawa, Japan; fukuzawa-tsuyoshi@tokai-u.jp; 2Muscle Physiology & Cell Biology Unit, Tokai University School of Medicine, 143 Shimokasuya, Isehara 259-1193, Kanagawa, Japan; nt554270@tsc.u-tokai.ac.jp (T.N.); 1cmud011@mail.u-tokai.ac.jp (M.T.); tokai-takeshi@tokai-u.jp (T.I.); ippei-y@is.icc.u-tokai.ac.jp (I.Y.); 3Department of Physiology, Tokai University School of Medicine, 143 Shimokasuya, Isehara 259-1193, Kanagawa, Japan; 4Department of Orthopedic Surgery, Tokai University School of Medicine, 143 Shimokasuya, Isehara 259-1193, Kanagawa, Japan; 5Department of Medical Education, Tokai University School of Medicine, 143 Shimokasuya, Isehara 259-1193, Kanagawa, Japan

**Keywords:** myogenic response, stem cells, satellite cells, proliferative capacity

## Abstract

The effects of total body irradiation (TBI) to the capacity of skeletal muscle hypertrophy were quantified using the compensatory muscle hypertrophy model. We additionally assessed the responses of stem and/or progenitor cells in the muscles. A single TBI of 9.0, 5.0 and 2.5 Gy was delivered to C57BL/6 mice. Bone marrow stromal cells were obtained from GFP-Tg mice, and were injected into the tail vein of the recipient mice (1 × 10^6^ cells/mouse), for bone marrow transplantation (BMT). Five weeks after TBI, the mean GFP-chimerism in the blood was 96 ± 0.8% in the 9 Gy, 83 ± 3.9% in the 5 Gy, and 8.4 ± 3.4% in the 2.5 Gy groups. This implied that the impact of 2.5 Gy is quite low and unavailable as the BMT treatment. Six weeks after the TBI/BMT procedure, muscle hypertrophy was induced in the right plantaris muscle by surgical ablation (SA) of the synergist muscles (gastrocnemius and soleus), and the contralateral left side was preserved as a control. The muscle hypertrophy capacity significantly decreased by 95% in the 9 Gy, 48% in the 5 Gy, and 36% in the 2.5 Gy groups. Furthermore, stem/progenitor cells in the muscle were enzymatically isolated and fractionated into non-sorted bulk cells, CD45-/34-/29+ (Sk-DN), and CD45-/34+ (Sk-34) cells, and myogenic capacity was confirmed by the presence of Pax7+ and MyoD+ cells in culture. Myogenic capacity also declined significantly in the Bulk and Sk-DN cell groups in all three TBI conditions, possibly implying that skeletal muscles are more susceptible to TBI than bone marrow. However, interstitial Sk-34 cells were insusceptible to TBI, retaining their myogenic/proliferative capacity.

## 1. Introduction

Total body irradiation (TBI) is an established treatment administered prior to allogeneic bone marrow transplantation (BMT), typically for leukemia, lymphoma, and other hematopoietic malignancies [1]. TBI exerts a myeloablative effect and promotes the engraftment of new donor hematopoietic stem cells. However, due to the nature of radiation (high-energy wave X-ray), proliferative stem and/or progenitor cells could be killed as a result of DNA damage. Although TBI targets the bone’s marrow, in order to suppress the immune system, it also induces several early and late side-effects in other organs such as the bone, brain, lung, ovary, kidneys, lens, and testis [2,3,4,5,6]. Although appropriate methods have been developed to protect the lungs and lens [1], it seems difficult to protect the other aforementioned organs. It also seems impossible to protect skeletal muscles, because they connect and situate around the bones as the body’s musculoskeletal framework and motor system. The musculoskeletal system is also known to have an increasing capacity for postnatal hypertrophy and regeneration. This indicates that stem/progenitor cells may be necessary to support these capacities, such as satellite cells [7,8], interstitial myogenic cells [9,10,11,12,13,14] and other peripheral nerve cells (Schwann, perineurial/endoneurial cells), and vascular cells (smooth muscle and endothelial cells, pericytes) [15]. However, these stem/progenitor cells in the skeletal muscles may also be affected by TBI. There are few reports on the detailed effects of TBI on the plasticity of skeletal muscles. For instance, a case of TBI treatment in childhood led to musculoskeletal complications, such as short stature [16]. In addition, a lifelong deficit in myofiber size and nuclear number with a loss of Pax7+ cells were induced by the prepubertal radiation to the skeletal muscle was also reported [17].

In the present study, a single TBI burst of 9.0, 5.0 and 2.5 Gy was used, and the bone marrow implantation (BMI) rate was determined to be an indicator of the effects of TBI. The percentage decrease in the capacity of skeletal muscle hypertrophy was quantified using the compensatory hypertrophy model in mice, which allowed us to obtain an almost maximum physiological hypertrophy rate [18,19]. Proliferation and differentiation capacity of the skeletal muscle stem/progenitor cells, such as Pax7+ putative satellite cells, were quantified. Furthermore, multipotent interstitial stem cells, capable of differentiating myogenic, vascular and peripheral nerve cells, were also sorted as skeletal muscle-derived Sk-34 (CD34+/45-) [12] and Sk-DN (CD34-/45-/29+) cells [11], and quantified and compared.

## 2. Methods

### 2.1. Experimental Animals

Green fluorescent protein-transgenic mice (GFP-Tg mice; C57BL/6 TgN[act EGFP]Osb Y01; produced by Dr. M. Okabe, Osaka University, Osaka, Japan) [20] were purchased (Japan LSC, Inc. Shizuoka, Japan) and used as donor mice for bone marrow transplantation (age 8–12 weeks, *n* = 18), and wild-type mice (C57BL/6N) were used as recipients (age 8–12 weeks, *n* = 76). All experimental procedures were approved by the Tokai University School of Medicine Committee on Animal Care and Use (no. 214015). All experiments were performed in accordance with relevant guidelines and regulations.

### 2.2. TBI and BMT

In the present study, we determined that 9.0 Gy was an appropriate maximum dose of TBI, as it was lethal but had a high recovery rate after BMT, and tolerated further surgical manipulation [21,22]. A single TBI burst of 9.0 Gy (*n* = 19), 5.0 Gy (*n* = 18), and 2.5 Gy (*n* = 18) was delivered to C57BL/6 mice using (HITACHI MBR-1520R-3, Tokyo, Japan). Simultaneously, whole bone marrow cells (BMCs) were obtained from the GFP-Tg mice (*n* = 18) by flashing the tibial and femoral bones. After eliminating red blood cells, freshly isolated BMCs were obtained and transplanted into the tail vein of each mouse (1 × 10^6^ cells/mouse) in the above three groups. Five weeks after BMT, the BMI rates were evaluated from blood samples taken from the tail vein. This was calculated as the replacement rate of GFP+ mononucleated cells (indicative of chimerism) in the peripheral blood, after dissolving the red blood cells in ammonium chloride solution (0.17 M NH_4_Cl, 10 mM Tris HCl, 0.25 mM EDTA) using FACSVerse (Nippon BD Co., Ltd., Tokyo, Japan). These values were taken individually and averaged as the group value.

The experimental protocol is shown in Figure 1. This study comprised of two main experiments; experiment 1 analyzed the effects of TBI on muscle hypertrophy capacity (Figure 1). Experiment 2 analyzed the effects of TBI on the stem/progenitor cells in skeletal muscles.

### 2.3. Compensatory Muscle Hypertrophy Model

After the confirmation of BMI rate five weeks after BMT, compensatory hypertrophy was induced in the 3 experimental groups as 9.0 Gy (*n* = 8), 5.0 Gy (*n* = 9), 2.5 Gy (*n* = 7) and the control group (*n* = 7). The right plantaris (PLT) muscle was chronically overloaded by surgical ablation (SA) of the synergistic muscles (gastrocnemius; GAS and soleus; SOL), and the contralateral left side was preserved as a control. Surgery was performed under inhalant anesthesia (Isoflurane; Abbott, Tokyo, Japan). The calf muscles were surgically exposed, and the hamstring was incised longitudinally and separated from the GAS muscle using blunt dissection. The distal tendons of the GAS and SOL muscles were isolated and sectioned. The SOL muscle was completely excised. The distal two-thirds of both the lateral and medial heads of the GAS muscle were clamped off with a hemostat and cut without bleeding, using an electrical knife. Care was taken during these procedures to not disturb the vessels and nerves in and around the remaining PLT muscle. The hamstring and skin were sutured with silk thread, and disinfectant plastic dressing (Nobecutane spray; Yoshitomi Chemical, Osaka, Japan) was applied. Penicillin was injected subcutaneously to prevent infection thereafter.

### 2.4. Evaluation of Muscle Hypertrophy Capacity and Isolation of Skeletal Muscle-Derived Cells

Five weeks after SA, animals in all four groups were anesthetized by sodium pentobarbital overdose (60 mg/kg, i.p.), and whole blood was taken by cardiopuncture. After confirming the BMI rate, the muscles were removed and weighed carefully. First, the operated and contralateral sides of the PLT were used for the individual evaluation of muscle hypertrophy. The remaining contralateral left side GAS and SOL muscles and both sides of the tibialis anterior (TA), extensor digitorum longus (EDL), and quadriceps femoris (QF) muscles were prepared for cell isolation, as described below. Non-SA mice with three kinds of TBI were also prepared for the following cellular analysis consistently used both sides of the TA, EDL, GAS, SOL, PLT and QF muscles (9 Gy: *n* = 11, 5 Gy: *n* = 9, 2.5 Gy: *n* = 11). As a result, we obtained the number of cells/animal following the specific determination by fluorescence-activated cell sorting (FACS) and/or immunocytochemistry. Note that in this study, in order to obtain total skeletal muscle-derived mononucleated cells with satellite cells, muscles were cut up into small pieces. This was different from our original isolation method, to prevent satellite cell contamination as much as possible [12]. The cut-up muscles were treated with 0.1% collagenase type IA (Sigma-Aldrich, St. Louis, MO, USA) in Dulbecco’s Modified Eagle’s Medium (DMEM), containing 7.5% fetal bovine serum (FBS), for 2 h at 37 °C with gentle agitation. This step allowed us to collect isolated cells along with satellite cells and other interstitial cells. The extracted cells were filtered through 70 µm, 40 µm and 20 µm nylon strainers to remove muscle fibers and other debris. They were then washed with DMEM and resuspended in Iscove’s Modified Dulbecco’s Medium (IMDM), containing 10% FBS, to protect enzymatically extracted cells. In this step, the enzymatically extracted mixed-cell population was established as the total bulk cells, and prepared for further analysis, as described below.

### 2.5. Fluorescence-Activated Cell Sorting (FACS) and Cell Culture

Three-quarters of the total bulk cells were prepared for FACS, in order to obtain Sk-DN (CD45-/34-/29+) and Sk-34 (CD45-/34+) cells (FACS Aria III, Nippon BD, Japan). Other cells were also counted. Anti-mouse CD29, CD45 (Biolegend, San Diego, CA, USA), and CD34 (BD Pharmingen, San Jose, CA, USA) were used for cell sorting. As a result, four unsorted groups of cells were obtained: unsorted bulk, Sk-DN, Sk-34, and other cells. The bulk, Sk-DN, and Sk-34 cells were then cultured in 20% FCS/IMDM for 5 days, and their differentiation/proliferation capacities were examined. After culturing, the cells were harvested using 0.25% Trypsin-EDTA (Life Technologies Japan, Tokyo, Japan), washed with DMEM, and further prepared for cytospin and mRNA analysis as described below.

### 2.6. Immunocytochemistry

For the cytospin preparation, suspended cells were fixed with 2% paraformaldehyde (PFA)/0.1 M phosphate buffer (PB) for 10 min, spun, and applied to a glass slide, and then re-fixed with 4% PFA/PB for 10 min and washed with 0.1 M phosphate buffered saline (PBS). The samples were then treated with 10–20% sucrose/PBS and frozen. After several repetitions of freezing/thawing, the samples were prepared for immunostaining, to analyze cell differentiation and proliferation capacity. Monoclonal anti-MyoD (myoblast determination protein 1, 1:50, 4 °C overnight; Dako, Carpinteria, CA, USA) antibodies were used to detect myogenic cells; monoclonal anti-Pax7 antibodies (Paired Box 7, 1:50, 4 °C overnight; Developmental Studies Hybridoma Bank, University Iowa, Iowa, IA, USA) were used to detect satellite cells; and monoclonal anti-PCNA antibodies (Proliferating Cell Nuclear Antigen, 1:1000, 4 °C overnight, Dako, Carpinteria, CA, USA) were used to detect proliferating cells.

Reactions were visualized using Alexa Fluor-594 conjugated goat anti-mouse antibodies (Molecular Probes, Eugene, OR, USA). Quantitative analysis was performed using the Stereo Investigator software (MBF Bioscience, Williston, VT, USA) and Photoshop (Adobe system Inc., San Jose, CA, USA). Data were averaged and expressed as percentages (positive cells/total cells).

### 2.7. Analysis of Specific mRNA Expression

Cellular differentiation capacity was also analyzed by the expression of specific mRNAs. In order to test the expression of specific markers for skeletal muscles, peripheral nerve and vascular cell lineages, and neurotrophic and vasculogenic factors, RT-PCR was performed on the cultured cells in four groups. Specific primers and analyzed materials are summarized in Appendix A. Cells were lysed and total RNA was purified using a QIAGEN RNeasy micro kit. First-strand cDNA synthesis was performed with an Invitrogen SuperScript III system using a dT30-containing primer, and 3 cycles of PCR (25, 28, and 30 cycles of 30 s at 94 °C, 30 s at 60–65 °C, and 2 min at 72 °C) were performed in a solution of 15 µL Ex-Taq buffer, 0.8 U of ExTaq-HS-polymerase, 0.7 µM specific sense and antisense primers, 0.2 mM dNTPs, and 0.5 µL of cDNA to be obtained relative expression intensity as the classification into 3 levels based on housekeeping control gene levels and averaged. Specific primers are summarized in Appendix A.

### 2.8. Statistics

Differences in all data among the four groups were analyzed using the parametric Tukey–Kramer post hoc test, and the significance level was set at *p* < 0.05. All values are expressed as mean ± SE.

## 3. Results

### 3.1. Body Mass, BMI Rate, and Muscle Hypertrophy Capacity

Final body mass, the BMI rate (% GFP+ cells; blood chimerism) and muscle hypertrophy capacity in the experimental groups and control group are shown in Figure 2A–C. Changes in body mass were unaffected by 2.5 and 5.0 Gy of TBI, but 9.0 Gy led to a significant decrease in body mass compared to the other three groups (Figure 2A). In contrast, a significantly higher BMI rate was observed in the 9.0 Gy group (96 ± 0.8%), indicating that this TBI intensity can successfully be used prior to BMT (Figure 2B). The 5.0 Gy group also showed a high BMI rate (83 ± 3.9%); thus, this intensity was also considered suitable prior to BMT treatment. However, the 2.5 Gy group showed lower BMI rate (8.4 ± 3.4%), indicating that this TBI intensity could not be used for BMT treatment.

A reverse trend of blood chimerism was observed in muscle hypertrophy capacity (Figure 2C). Significant decline in this capacity was observed in the 9.0 Gy condition (5.3 ± 5.1%), to the extent that there remained almost no hypertrophy capability. However, the 5.0 Gy and 2.5 Gy conditions showed a relatively higher capacity (51.5 ± 2.6% and 74.1 ± 8.3%), while all three groups had a significant reduction in muscle hypertrophy capacity of over 25%. Therefore, these findings likely revealed a dose-dependent relationship.

The effects of TBI on body and contralateral muscle mass are presented in Table 1. The PLT and TA muscles were selected as typical plantar flexor and dorsiflexor muscles. The contralateral muscles of all three groups consistently showed a downward trend in their mass. Therefore, TBI was likely to have a greater impact on skeletal muscle mass than body mass. However, a significant decrease was found only in the 9 Gy TA muscle.

### 3.2. Total Enzymatically Isolated Cells and Their Distribution in Sk-DN, Sk-34, and Other Cells

In order to further examine the physiological events related to the significant decrease in muscle hypertrophy capacity, all mononucleated cells in skeletal muscles were enzymatically isolated and characterized. First, total isolated live cells were counted (Figure 3A) and then sorted by FACS into three groups: Sk-DN, Sk-34, and other cells (Figure 3B–G). The total number of isolated cells was normalized per one animal value. As expected, there was a significant decrease compared to the control group consistently in three TBI groups. Additionally, there were no significant differences between the three TBI groups (Figure 3A). Similarly, a significant decrease in the number of Sk-DN, Sk-34, and the other cells was seen in Figure 3B–D. However, the percentage distribution of Sk-DN and Sk-34 cells was similar between the control and TBI groups (Figure 3E,F). Furthermore, the percentage inclusion of other cell types relatively increased with lower TBI intensity; a significant increase was seen in the 2.5 Gy TBI group compared to the control group (Figure 3F). There was additionally no dose-dependence relationship underpinning this observation.

### 3.3. Cellular Proliferation and Differentiation Capacity in the Muscles

Bulk cells were cultured for 5 days, following which cellular proliferation and myogenic differentiation capacity were analyzed using cytospin protocol. Total cell count imaging using DAPI staining was conducted, as shown in Figure 4A, and positive/negative identification is presented in Figure 4B. Pax7+, MyoD+, and PCNA+ cells were additionally counted and compared. The presence of Pax7+ cells indicated putative satellite cells (Figure 4C). The control group had a significantly higher number of Pax7+ cells (8.3 ± 1.8%) than the 9.0 Gy (0.2 ± 0.1%), 5.0 Gy (1.2 ± 0.4%), and 2.5 Gy (2.9 ± 0.7%) groups. Myogenic differentiation capacity was also assessed by the presence of MyoD+ cells (Figure 4D). The observed cellular responses were identical to those from Pax7+ cells. A significant decrease in the number of MyoD+ cells was observed in all three TBI groups compared to the control group, similar to Pax7 expression. In addition, significant differences were observed among the TBI groups (Figure 4D). On the other hand, no significant differences were observed in the presence of PCNA+ cells between the control and TBI groups (Figure 4E). This indicated that the cellular proliferation capacity of the skeletal muscles was not affected by TBI totally. In addition, some myogenic cells (Pax7+ and/or MyoD+ cells) may have also shown proliferative capacity and have been included in the PCNA+ cell population, but these were significantly decreased (Figure 4C,D). Therefore, it could be presumed that a different cell type, other than myogenic cells, may have contributed to proliferation capacity.

### 3.4. Putative Cellular Differentiation Capacity in Non-Myogenic Cells

As per the above results, TBI induced a significant reduction in the capacity of muscle hypertrophy. Although significant decrease of the myogenic cells was found, cellular proliferation capacity did not change. Some myogenic cells may have been included in the PCNA+ cell population, but it was unclear whether other cell types were present in these proliferative cells. We thus performed RT-PCR analysis to detect together the markers of myogenic cells (red; 1–7), fibrosis (green; 8–13), peripheral nerve (pink; 14–20), and vasculogenic cells (blue; 21–23) (Figure 5). Expressions of mRNAs were represented relative intensity, which was determined by three graded PCR cycles. First, the expression of myogenic markers (red) reduced following an increase in TBI intensity compared to the control group; this was identical to our previous immunocytochemistry results (Figure 4). Second, a gradual increase in fibrosis-related markers (green) was evident; this was inverse to the expression of myogenic markers. Third, the expression of peripheral nerve and vascular-related markers did not change. This result strongly suggested that fibroblastic cell population could be induced higher number of PCNA+ cells.

### 3.5. Cellular Proliferation and Differentiation Capacity of the Sk-DN and Sk-34 Cells

Since the enzymatically extracted bulk cells contained skeletal muscle-derived Sk-DN and Sk-34 stem cells [11,12], these cells were cultured for 5 days, after which immunocytochemistry and cytospin analyses were performed (Figure 6). The culture conditions were the same as previously described for the bulk cells.

A similar trend with the bulk cells was found in the Sk-DN cells (Figure 6A–C). Percentage of Pax7+ cells drastically increased in the control (30.0 ± 1.4%), 5.0 Gy (9.3 ± 0.1%), and 2.5 Gy (22.7 ± 3.6%) groups, respectively (Figure 6A). These increases were evaluated to be 3.6–7.6-fold against the case of Bulk cells, indicating enrichment of Pax7+ cells. This trend of enrichment is also detected in the MyoD+ cells (Figure 6B). However, it should be noted that Pax7+ and MyoD+ cells could not be detected in the 9.0 Gy condition, and this trend was also observed in the Sk-34 cells, indicating a strong impact of 9.0 Gy. Additionally, only a small number of Pax7+ and MyoD+ cells could be detected in Sk-34 cells in all conditions (Figure 6D,E), indicating fewer myogenic cells in the freshly isolated Sk-34 cell fraction.

Myogenic differentiation capacity was analyzed by the presence of MyoD+ cells (Figure 6B,E). It was observed that cellular responses were identical to those from Pax7+ cells. Myogenic MyoD+ cells were also increased in the Sk-DN cell fraction, and significant differences were found between the control and TBI group (Figure 6B). We also found decreased myogenic cells in the Sk-34 group (Figure 6E). On the other hand, no significant differences between the control and TBI groups were observed in the presence of PCNA+ cells (Figure 6C,F). Therefore, it was considered that TBI did not exert strong effects on cellular proliferative capacity, except for the 9.0 Gy condition. Relatively higher values were found in the Sk-34 cells of the control group (Figure 6F).

Thus, the Sk-DN cell fraction were found to be myogenic cells enriched fraction that were susceptible to the effects of TBI, whereas the Sk-34 cells were proliferative less-myogenic cells that were insusceptible to the effects of TBI. Additionally, it should be noted that the Sk-34 cell fraction was of the main cellular component of the enzymatically isolated cells, and the absolute value of this fraction was 5-fold higher than the Sk-DN cell fraction (see Figure 3).

## 4. Discussion

The present study indicates that cell/tissue responses of the bone marrow and skeletal muscles differ under three graded TBI intensities. At a lethal high intensity TBI of 9 Gy, successive BMT was achieved with an average replacement of 96% of donor GFP+ cells. The proliferative capacity of stem/progenitor cells in the bone marrow may have therefore ceased to multiply. Similarly, the capacity of skeletal muscle hypertrophy was disabled by 97–98%, inversely parallel to the bone marrow. In addition, the mice in the 9.0 Gy group also had a significantly decreased final body mass (Figure 2) and TA muscle mass (Table 1), showing that a direct effect of TBI on the sedentary muscle occurred without specific stimulation. Therefore, a TBI dose of 9.0 Gy affected the entire body. Changes in blood chimerism and muscle hypertrophy ratio likely depended on dose, but it can be assumed that a threshold exists between 5.0 and 2.5 Gy. However, a TBI dose of 2.5 Gy had no effect on bone marrow, but skeletal muscles showed significant decrease in muscle hypertrophy capacity (>25%; Figure 2). In this aspect, bone morrow and skeletal muscle responses appear to be different, and the latter may be somewhat sensitive to TBI.

Decreased muscle hypertrophy capacity was also reflected on the cellular level in the skeletal muscles, as a significantly decreased number of cells could be isolated per animal (Figure 3A). The number of total cells and the three cell groups consistently showed significant a decrease in the three TBI intensities (Figure 3A–D), with no dose-dependence relationship. Additionally, the distribution of Sk-DN and Sk-34 cells was not affected by TBI, but other cell types increased significantly after 2.5 Gy TBI (Figure 3G). A similar increasing trend was observed in the ratio of PCNA+ cells, whereas myogenic cells decreased significantly (Figure 4C–E). In addition, the mRNA analysis suggested muscle fibrosis after TBI (Figure 5). Therefore, the decrease in CBT-induced myogenic potential and increase in skeletal muscle fibrosis, indicates the increased proliferation capacity of fibroblasts. Radiation-induced acceleration of fibrosis has been reported frequently in human clinical cases in various tissues, such as the lungs [6,23], kidneys [24], and skin [25]. This has also been observed previously in animal models [26,27,28]. Therefore, the current results correspond to these reports and also suggest that fibrosis is typical occurrence after TBI.

It should be noted that in order to obtain mononuclear cells derived from skeletal muscles with satellite cells in this study, the muscles were cut into small pieces prior to collagenase treatment. Therefore, the total isolated cells also included satellite cells [7,29,30] and other interstitial stem/progenitor cells [10,11,12,13]. These interstitial stem cells are also myogenic; therefore, it is unclear which of the cells displayed and declined in myogenic potential. The ratio of myogenic cells in the bulk (Figure 4C,D) and Sk-DN (Figure 6A,B) cell groups were significantly decreased, but Sk-34 cells were not affected by TBI (Figure 6D,E). It could therefore be considered that the myogenic Sk-34 cells had restored their myogenic ability. Importantly, as observed in Figure 6, there were likely to be a few myogenic Sk-34 cells; however, it should be noted that Sk-34 cells were the main cellular component of the enzymatically isolated cells, and the total number of Sk-34 cells was 5-fold higher than that of the Sk-DN cells (see Figure 3). Our previous studies found that satellite cells were CD34- [12,31,32], and that only a few Pax7+ putative satellite cells were included in the Sk-DN fraction in freshly enzymatically isolated cells [33]. In the present study, it should be considered that the Sk-DN cell fraction was composed of a mix of satellite cells and Sk-DN cells. Satellite cells are located between the basal lamina and plasma membrane of the muscle fibers; however, the Sk-DN cells are located in the interstitial spaces. Therefore, the myogenic capacity of the interstitial Sk-DN and Sk-34 cells could be restored. However, a significant decrease in the myogenic cell population was detected in the Sk-DN cell fraction (Figure 6). Thus, it is assumed that myogenic cells differ from the original Sk-DN cells, which might be mainly affected by TBI, such as satellite cells. In this regard, radiation-induced depletion of satellite cells has also been previously reported [17,34,35]. It is also hypothesized currently that the differences in cellular location, such as in between the basal lamina and plasma membrane of muscle fibers, or in the interstitial connective tissue network, could possibly be related to resistance against radiation. Further research is needed to be ascertain this.

In the clinical aspect, it is important to consider the impact of TBI-related side effects on skeletal muscles, because of this close relationship to general physical activity and quality of life. For this, we need to quantify the impact of general activity to the skeletal muscles in daily life, based on their maximum physiological capacity. In a previous study, muscle hypertrophic effects in a rat model were estimated as amino acid uptake per unit time using [^14^C]leucine (disintegration per minute; dpm/mg), while performing an exhaustive weight-lifting (squat) exercise; this study showed 3.3–3.5-fold higher hypertrophic effects than the resting level after 2–4 days [36]. This uptake was then accelerated to approximately 5-fold the resting level by the administration of anabolic androgenic steroids (AAS) [37]. A subsequent compensatory hypertrophy SA model in rats showed an approximately 10-fold increase in hypertrophic capacity after 3–5 days, and this was not accelerated by AAS administration [19]. This implies that the SA model could almost induce the maximum physiological hypertrophy rate [19]. Based on these data, it can be hypothesized that we usually use <30% of the maximum capacity of our skeletal muscles in daily life or after physical exercise, and the aforementioned study also showed that the side effects of a 5.0 Gy TBI dose (decreased to approximately 50%) is of allowable range.

Similar side effects may occur in other organs and tissues, and to this end, fractionated irradiation methods have been used clinically [2,4] and experimentally evaluated [38] to reduce side effects. Although we did not use the fractionated irradiation method in the current study, based on the present results, we posit that the effects of 3 × 2.5–3.0 Gy fractioned irradiation treatment should be further researched to observe its effects on skeletal muscles.

Finally, we did not address the analysis of the main parameters of skeletal muscle atrophy such as the E3 ubiquitin ligases MuRF1 and MAFbx/atrogin-1 [39,40,41] in this study, because an apparent trend of muscle atrophy, such as the decline of the muscle/body mass index, could not be detected in three groups during a current experimental period. However, we also consider the relationship between ubiquitin ligases and radiation to be an important issue. Therefore, we would analyze these issues as the next step in relation to the longer-term effects of muscle radiation.

## Figures and Tables

**Figure 1 jcm-11-03735-f001:**
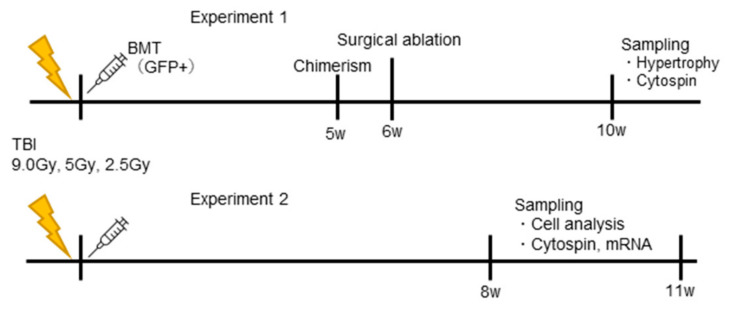
Experimental protocol. Experiment 1: Analysis of muscle hypertrophy. Experiment 2: Analysis of myogenic and proliferative capacity of the skeletal muscle-derived stem/progenitor cells.

**Figure 2 jcm-11-03735-f002:**
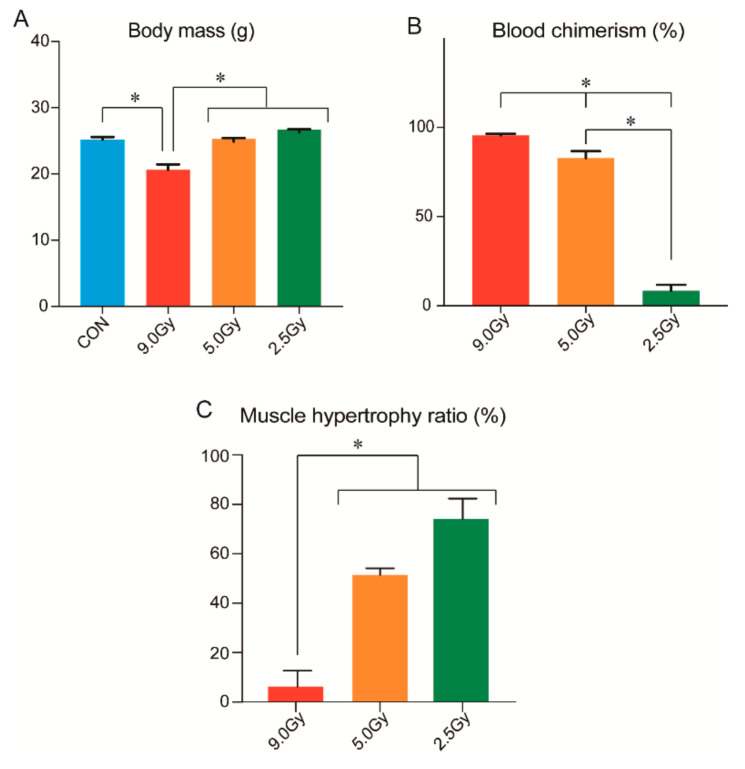
Body mass, BMI rate, and muscle hypertrophy capacity in the three TBI groups. (**A**): Significant decrease in body mass was only present in the 9.0 Gy group compared to the other three groups. (**B**): GFP+ peripheral blood, indicative of chimerism and mononucleated cells. Almost no TBI effects were observed in the 2.5 Gy group. However, significant differences were observed in each three groups. (**C**): Muscle hypertrophy capacity. Values were set at 100% of the normal control value. CON = control group; * *p* < 0.05.

**Figure 3 jcm-11-03735-f003:**
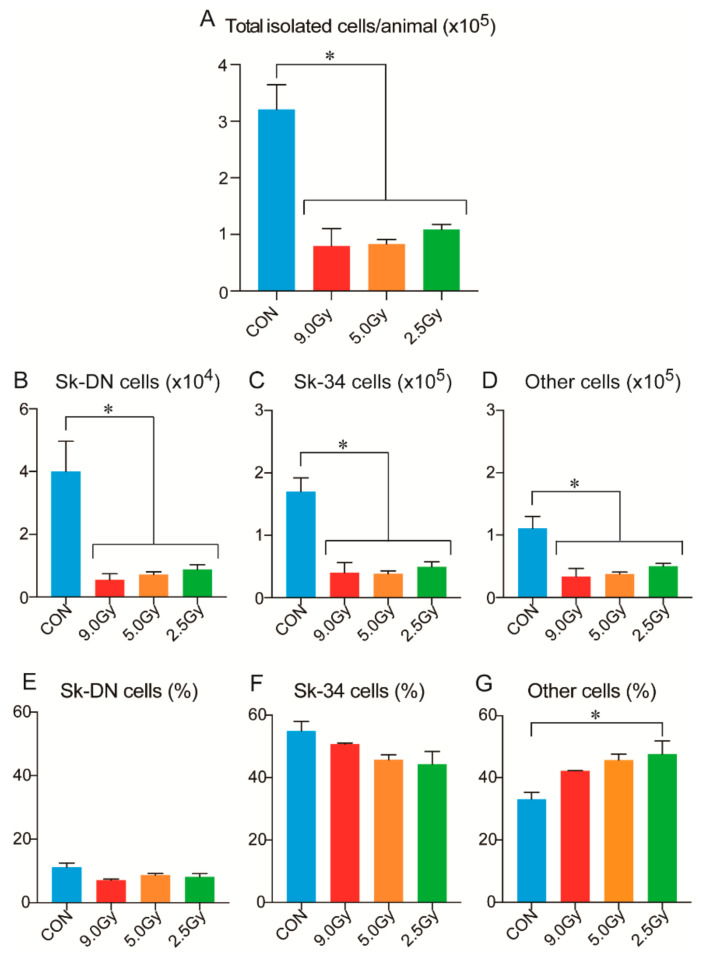
The number of enzymatically isolated cells and Sk-DN, Sk-34 and other cells. All values were normalized per one animal. Both sides of the TA, EDL, GAS, SOL, PLT and QF muscles were consistently used. (**A**) Absolute value of isolated live cells. (**B**) Absolute value of Sk-DN cells. (**C**) Absolute value of Sk-34 cells. (**D**) Absolute value of other cells. (**E**) Percentage of Sk-DN cells. (**F**) Percentage of Sk-34 cells. (**G**) Percentage of other cells. CON: normal control group. Sk-DN = CD45-/34-/29+, Sk-34 = CD45-/34+, other cells=all marker negative. Values are expressed as mean ± SE. * *p* < 0.05.

**Figure 4 jcm-11-03735-f004:**
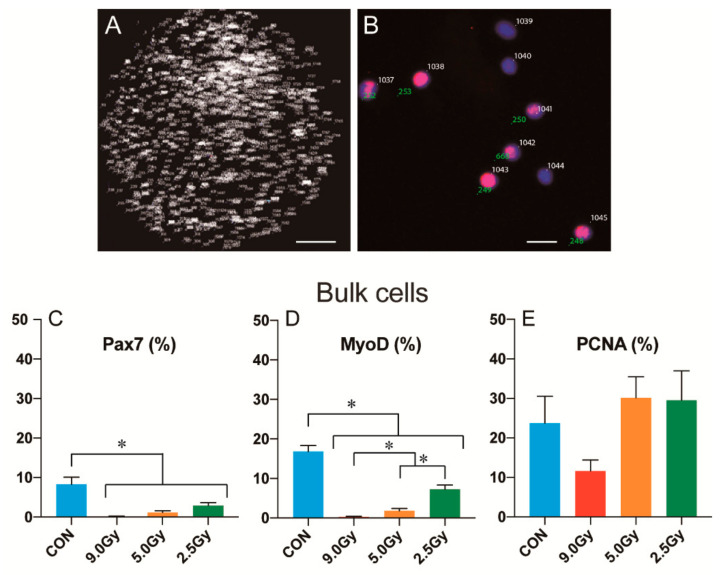
Quantitative analysis using immunocytochemistry. (**A**) Total cell count for bulk cells, depicted via a tiled image of high magnification (100×) photographs (64 pictures). Count numbers are provided next to the cells. Cell counting was used DAPI staining (blue staining, see panel (**B**)), but this image is in black and white, due to the count number display. The scale bar represents 1 mm. (**B**) High-magnification image of MyoD+/− cells, and identification of these counting. Pink staining is MyoD+ cells. The numbers entered in the upper right indicate the part of total cell count (white), and those on the lower left indicate MyoD+ cells (green). The scale bar represents 20 μm. (**C**) The distribution ratio of Pax7+ cells in the bulk cells. (**D**) The distribution ratio of MyoD+ cells. (**E**) The distribution ratio of PCNA+ cells. Values were calculated as percentages of positive cells/total cells and expressed as mean ± SE. * *p* < 0.05.

**Figure 5 jcm-11-03735-f005:**
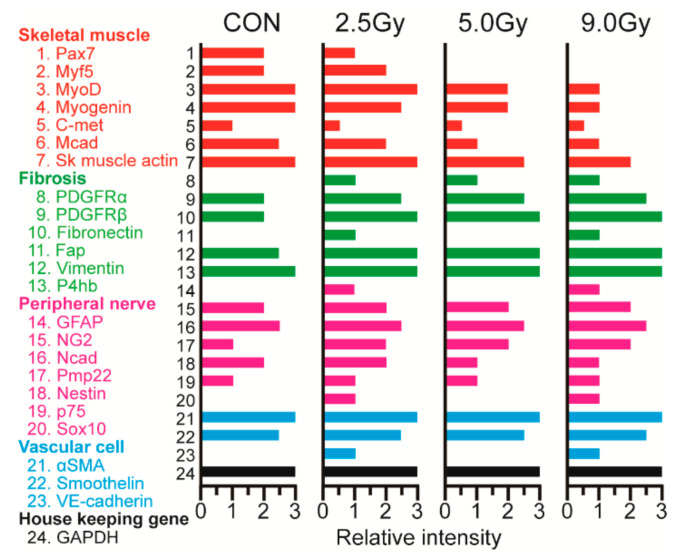
Relative expression of myogenic (red; 1–7), fibrosis-related (green; 8–13), peripheral nerve-related (pink; 14–20), and vasculogenic (blue; 21–23) mRNAs in bulk cells. The relative expression of these markers was determined by the strength of the band on 25, 28, and 30 graded PCR cycles, based on the housekeeping gene.

**Figure 6 jcm-11-03735-f006:**
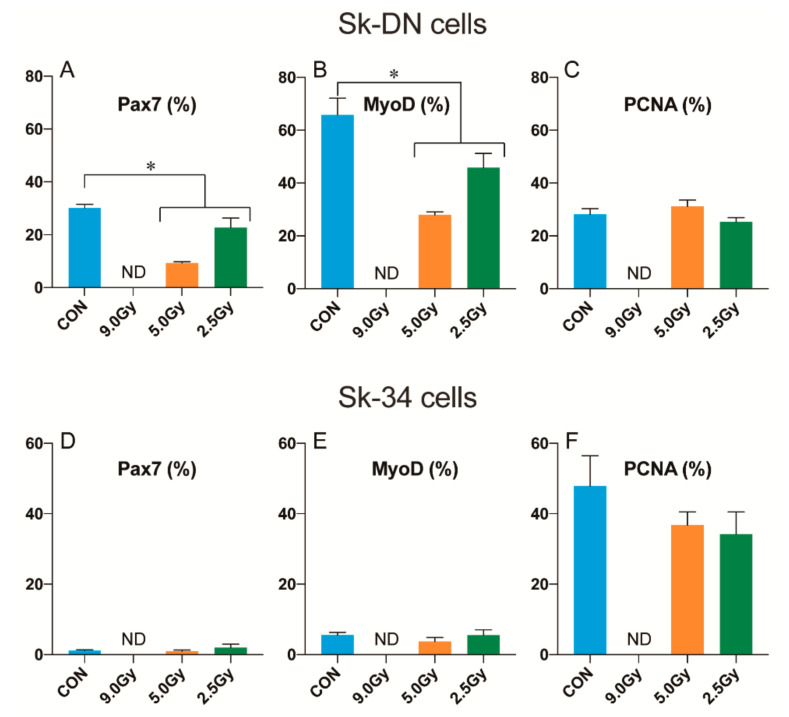
Myogenic differentiation and proliferation capacity in Sk-DN and Sk-34 cells isolated from normal and TBI muscles. Cells were sorted as CD45-/34-/29+ fraction ((**A**–**C**), Sk-DN) and CD45-/34+ group ((**D**–**F**), Sk-34), and cultured for 5 days. Myogenic differentiation capacity was examined by Pax7 and MyoD expression, and cellular proliferation capacity was determined by PCNA expression. Values were calculated as the percentage of positive cells/total cells, and expressed as mean ± S.E. * *p* < 0.05.

**Table 1 jcm-11-03735-t001:** Basic data of body and muscle mass after TBI.

	Body Mass (g)		Contralateral Muscle Mass (mg)	
	Before TBI	10 Weeks after TBI	Plantar Flexor PLT	Dorsiflexor TA
9.0 Gy	20.6 ± 0.4	19.4 ± 0.9 *^#^^φ^	14.4 ± 0.7	38.2 ± 1.9 *
5.0 Gy	20.7 ± 0.4	26.2 ± 0.7	14.0 ± 0.5	44.6 ± 1.8
2.5 Gy	19.6 ± 0.3	26.4 ± 0.5	14.3 ± 0.6	41.0 ± 1.4
CON	20.6 ± 0.5	25.0 ± 0.6	15.2 ± 0.1	49.1 ± 0.8

* 9.0 Gy vs. CON, ^#^ 9.0 Gy vs. 5.0 Gy, ^φ^ 9.0 Gy vs. 2.5 Gy.

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
