# Peer review of "Quantitative Evaluation of the Reduced Capacity of Skeletal Muscle Hypertrophy after Total Body Irradiation in Relation to Stem/Progenitor Cells"

_jcm, 2022, doi:10.3390/jcm11133735_

Round 1

Reviewer 1 Report

In this study, Fukuzawa et al. investigate the effect of total body irradiation on stem/progenitor cells in skeletal muscle. They use 3 different irradiation doses and monitor the effect on hypertrophy, cell number and proliferation. The authors find that the highest irradiation dose results in an almost full replacement of bone marrow transplantation but drastic reduction in hypertrophy capacity and myogenic progenitor numbers. Although the work displays some convincing results, the manuscript could be strengthened on both the writing and experimental front.

Major points:

1. The introduction should be lengthened to better contextualize the study. A. To provide importance to the study, clinical manifestations of TBI on the musculoskeletal system should be highlighted and discussed. Other than one case of short stature, the authors do not highlight why TBI is a concrete problem for muscle biology. B. The authors could delve a bit more into studies that explore the effect of radiation on muscle stem cells (activation, proliferation and differentiation). These can enlighten the reader on the cellular consequences of TBI on satellite cells and provide good foundations by which to introduce their work. C. A better description of the SK-DN and SK-34 populations and the choice for isolating these specific populations would help the reader understand the author’s research approach.

2. The results section could use more context. For example, the results section begins directly with the reference to figure 2 (in the first sentence). Introducing hypotheses, rationale and experimental approach would help the reader understand why and how the authors are performing experiments.

3. Figure 4E does not display statistics. The authors claim that there is no difference between control and TBI groups but levels of PCNA positive cells differ greatly, especially for the 9.0GY dose. Other proliferation assays should be considered to strengthen the conclusions here.

4. In figure 5, the authors rely on RT-PCR data to extrapolate the changes in cell populations after TBI. Although this section ends with a suggestive conclusion, RT-PCR assays are not a good experimental approach to conclude on population numbers.

Minor points:

1. Please describe some technical terms such as PCNA+, Pax7 or MyoD. Although the reader has a sense of what these are, a clear description of why these markers are tested would be helpful.

2. Figure 5 is wrongly referenced as Figure 6.

3. PCNA+ representative image should be provided to give the reader an idea of what this proliferation assay looks like.

4. The axis of many plots could be changed to better see the data. For example, in figure 4 C-E, the authors try to maintain consistency across markers with the y axis set at maximum 50% value but 20% would be more appropriate to gauge percent differences of Pax7 expressing cells.

Author Response

Point-by-point responses

For the reviewer 1.

We thank the reviewer for a favorable reading and helpful comments.

Major points:

  1. The introduction should be lengthened to better contextualize the study. A. To provide importance to the study, clinical manifestations of TBI on the musculoskeletal system should be highlighted and discussed. Other than one case of short stature, the authors do not highlight why TBI is a concrete problem for muscle biology. B. The authors could delve a bit more into studies that explore the effect of radiation on muscle stem cells (activation, proliferation and differentiation). These can enlighten the reader on the cellular consequences of TBI on satellite cells and provide good foundations by which to introduce their work. C. A better description of the SK-DN and SK-34 populations and the choice for isolating these specific populations would help the reader understand the author’s research approach.

Response:

To point A:

First of all, we think that exposure of radiation to the skeletal muscle is an unavoidable side effect of TBI, and this has been insufficiently studied. In this respect, we believe we provide enough context in the introduction.  

To point B:

We thank the reviewer for the valuable suggestion. We therefore added a reference, which described “Radiation-Induced Damage to Prepubertal Pax7+ Skeletal Muscle Stem Cells Drives Lifelong Deficits in Myofiber Size and Nuclear Number” at the end of the first paragraph in the Introduction.

To point C:

We appreciate your focus on Sk-34 and Sk-DN cells. Both cells are our originally isolated/established cells exited interstitial spaces of skeletal muscle consistently across the mouse and human. We added more information in the last paragraph of the Introduction.    

  1. The results section could use more context. For example, the results section begins directly with the reference to figure 2 (in the first sentence). Introducing hypotheses, rationale and experimental approach would help the reader understand why and how the authors are performing experiments.

Response:

We are not sure we know what the reviewer means. However, we believe that the result section should provide a concise and precise description of the experimental results, their interpretation as well as the experimental conclusions that can be drawn.

  1. Figure 4E does not display statistics. The authors claim that there is no difference between control and TBI groups but levels of PCNA positive cells differ greatly, especially for the 9.0GY dose. Other proliferation assays should be considered to strengthen the conclusions here.

Response:

On Figure 4E, as suggested by the reviewer, the level of PCNA+ cells differ greatly between 9 Gy group, and the other three groups, at a glance. However, there is no significant difference in output can in fact be due to the great SD in each group. We have totally/strictly analyzed all data by the same standard (Tukey’s multiple comparisons test). Therefore, we cannot help but say that "there is no difference between control and the other TBI groups".  

  1. In figure 5, the authors rely on RT-PCR data to extrapolate the changes in cell populations after TBI. Although this section ends with a suggestive conclusion, RT-PCR assays are not a good experimental approach to conclude on population numbers.

Response:

Yes, the reviewer is right. The RT-PCR findings are not conclusive, thus, we clearly stated as in “3.4. Putative cellular differentiation capacity in non-myogenic cells”. Therefore, our conclusion is only a suggestion after the comparison of relative expressions of the other myogenic markers, which could obtain the numbers preliminary (Figs, 4 and 6). In fact, there is no suitable antibody marker for fibroblast. However, we could carry out a relative comparison, then suggested that the relative increase in fibroblasts markers mRNAs might be reflect the number of PCNA+ (proliferating) cells.

Minor points:

  1. Please describe some technical terms such as PCNA+, Pax7 or MyoD. Although the reader has a sense of what these are, a clear description of why these markers are tested would be helpful.

Response:

There were shortly described in Method 2.6. Immunocytochemistry. There are not so specific markers in the area of stem cells. We also spelt out these markers, representing typical role of these nuclear proteins.  

 Figure 5 is wrongly referenced as Figure 6.

Response:

We sincerely thank the reviewer. It was an incorrect description, and we corrected it. (Line 269)

  1. PCNA+ representative image should be provided to give the reader an idea of what this proliferation assay looks like.

Response:

                   Again, we thank the reviewer’s kind suggestion. However, the markers currently used, MoyD, Pax7 and PCNAs are all nuclear proteins. MyoD is a factor of myogenic regulatory factors (MRF) Myf5, MyoD, myogenin and MRF4, which are members of the helix-loop-helix family of transcription factors. Similarly, Pax7 belongs to the paired box protein family of transcription factors, which is an important regulator of neural and skeletal muscle development. Then, PCNA is a proliferating cell nuclear antigen. As a result, the images of these makers are completely the same as presented in Fig. 4B (positive=pink nuclear, negative=blue; DAPI only). These markers are very common in stem cell studies, particularly in stem cells derived from muscle. Therefore, we think that the morphological picture may be sufficient with Fig. 4B. 

  1. The axis of many plots could be changed to better see the data. For example, in figure 4 C-E, the authors try to maintain consistency across markers with the y axis set at maximum 50% value but 20% would be more appropriate to gauge percent differences of Pax7 expressing cells.

Response:

We appreciate your suggestion, which is a legitimate concern. However, we would like to represent any change in all data such as image set with uniform intent, for easy comparison across data. Additionally, small numerical values are described individually in the text of the results section as needed.

Reviewer 2 Report

The authors summarized the effect of Total Body Irradiation on Skeletal Muscle and Stem/Progenitor cells in state-of-the-art manner. Authors discussed the effects of TBI on different parameters, however, they didn’t establish any link between TBI-induced atrophy and attenuated myogenesis. It is expected the discuss how TBI induces skeletal muscle atrophy and depleted Stem/Progenitor cells with appropriate references. DNA fragmentation is one factor to induce atrophy along with other factors. Moreover, the authors did not study major parameters of skeletal muscle atrophy like Atrogin, Murf, etc., hence authors should discuss the mechanism of TBI induced atrophy apart from DNA fragmentation at least in few lines with references.

Author Response

Point-by-point responses

For the reviewer 2.

We thank the reviewer for a favorable reading and helpful comments. The reviewer is correct, and we did not examine the main parameters of skeletal muscle atrophy such as the E3 ubiquitin ligases Atrogin-1 and Murf-1 in this study. The reason is that we currently focus the capacity of muscle hypertrophy under three different doses of TBI. In addition, muscle atrophy has not occurred under the index of calculation muscle mass/body mass (the 9 Gy group showed the highest values of index as 1.97=TA and 0.74=PLT rather than the others). However, the reviewer is right, and we also think that the relationship between the ubiquitin ligase and the radiation is an important issue. Therefore, we added one paragraph at the end of the Discussion as follows. 

Finally, we did not address the analysis of the main parameters of skeletal muscle atrophy such as the E3 ubiquitin ligases MuRF1 and MAFbx/atrogin-1[39,40,41] in this study, because an apparent trend of muscle atrophy, such as the decline of the muscle/body mass index, could not be detected in three groups during a current experimental period. However, we also consider the relationship between ubiquitin ligases and radiation to be an important issue. Therefore, we would analyze these issues as the next step in relation to the longer-term effects of muscle radiation.

Round 2

Reviewer 1 Report

Thank you for taking into consideration the reviews.